# Pandemic-associated mobility restrictions could cause increases in dengue virus transmission

**Sean M. Cavany**[1]*, **Guido España**[1], **Gonzalo M. Vazquez-Prokopec**[2], **Thomas W. Scott**[3], **T Alex Perkins**[1]*

**1** Department of Biological Sciences, University of Notre Dame, Notre Dame, Indiana, United States of America, **2** Department of Environmental Sciences, Emory University, Atlanta, Georgia, United States of America, **3** Department of Entomology and Nematology, University of California, Davis, Davis, California, United States of America

* scavany@nd.edu (SMC); taperkins@nd.edu (TAP)

## Abstract

### Background

The COVID-19 pandemic has induced unprecedented reductions in human mobility and social contacts throughout the world. Because dengue virus (DENV) transmission is strongly driven by human mobility, behavioral changes associated with the pandemic have been hypothesized to impact dengue incidence. By discouraging human contact, COVID-19 control measures have also disrupted dengue vector control interventions, the most effective of which require entry into homes. We sought to investigate how and why dengue incidence could differ under a lockdown scenario with a proportion of the population sheltered at home.

### Methodology & principal findings

We used an agent-based model with a realistic treatment of human mobility and vector control. We found that a lockdown in which 70% of the population sheltered at home and which occurred in a season when a new serotype invaded could lead to a small average increase in cumulative DENV infections of up to 10%, depending on the time of year lockdown occurred. Lockdown had a more pronounced effect on the spatial distribution of DENV infections, with higher incidence under lockdown in regions with higher mosquito abundance. Transmission was also more focused in homes following lockdown. The proportion of people infected in their own home rose from 54% under normal conditions to 66% under lockdown, and the household secondary attack rate rose from 0.109 to 0.128, a 17% increase. When we considered that lockdown measures could disrupt regular, city-wide vector control campaigns, the increase in incidence was more pronounced than with lockdown alone, especially if lockdown occurred at the optimal time for vector control.

### Conclusions & significance

Our results indicate that an unintended outcome of lockdown measures may be to adversely alter the epidemiology of dengue. This observation has important implications for an

**Data Availability Statement:** All model code and synthetic data are available at github.com/scavany/dengue_shelter_in_place.

**Funding:** All authors were supported by grant P01AI098670 (TWS, PI) from the National Institutes of Health, National Institute for Allergy and Infectious Disease (https://www.niaid.nih.gov). GE and TAP were also supported by a RAPID grant from the National Science Foundation (DEB 2027718). The funders had no role in the study design, data collection and analysis, decision to publish, or preparation of the manuscript.

**Competing interests:** The authors have declared that no competing interests exist.

improved understanding of dengue epidemiology and effective application of dengue vector control. When coordinating public health responses during a syndemic, it is important to monitor multiple infections and understand that an intervention against one disease may exacerbate another.

## Author summary

Dengue virus causes substantial suffering in the tropical and subtropical regions of the world, with roughly 400 million infections and 40,000 deaths each year. In 2020, we witnessed unprecedented changes in human movement as the world tried to combat the COVID-19 pandemic, including in countries that regularly experience dengue epidemics, such as Thailand and Peru. These changes could affect transmission of dengue virus, though it is unclear whether transmission will decrease, as people reduce their movements between houses, or increase, as people spend more time at home and campaigns to control the mosquito vector of dengue virus are interrupted. We used a simulation model to estimate the impact of these changes on dengue virus transmission. Our model describes the locations of buildings and the movement of people between them, allowing us to directly estimate what happens when human movement patterns change. We found that as people spend more time at home, transmission is likely to increase moderately. If these changes also lead to disruption to vector control, the magnitude of the increase is greater. Our results reinforce concerns about the complexity of public health responses to multiple overlapping epidemics and support the need for policy makers and health authorities to think holistically in their intervention planning.

## Introduction

Interventions to combat the COVID-19 pandemic have had unprecedented effects on the lives of people around the world. While measures like social distancing and stay-at-home orders have been successful in reducing transmission, morbidity, and mortality associated with SARS-CoV-2 [1], they are likely to have also had an effect on the incidence of other diseases. For example, lockdown measures are predicted to increase the burden of tuberculosis, HIV, and malaria by reducing access to essential services [2,3].

Dengue is a mosquito-borne viral disease endemic across much of the tropics, with an estimated 400 million infections and 40,000 deaths each year [4,5]. A number of countries with particularly severe COVID-19 epidemics regularly experience dengue epidemics (e.g., Peru, Brazil, and Indonesia), and there have been reports of 2020 being an above-average year for dengue in South America [6]. A number of warnings have been raised regarding the potential dangers of overlapping dengue and COVID-19 epidemics; e.g., both diseases can result in similar symptoms and there have been reports of serological cross-reaction, which increases the chance of misdiagnosis [7–13]. At least five cases of dengue-COVID-19 co-infection have been reported, one of which resulted in death by stroke [8,13–17]. Some researchers have raised concerns about the possible impact of interrupted vector control campaigns and called for efforts to overcome this adverse impact [18,19].

Dengue virus (DENV) transmission is influenced by multiple overlapping drivers, including human and mosquito movement; climate and environmental factors that affect mosquito abundance, contact with human hosts, and vector-virus interactions; human host immunity;

and virus genotype [20–25]. Hence, predicting the epidemiological impact of drastic changes in human mobility on DENV transmission is difficult because of the large number of other potentially influential variables that interact in complex ways. It is understood that daily routine movement between houses is a key driver of DENV transmission, and it is this type of movement that has been most affected by COVID-19 lockdown measures [23]. While on the one hand reductions in house-to-house movements could be expected to reduce DENV transmission, on the other hand increased opportunities for intra-household transmission coupled with local mosquito movement and imperfect compliance with lockdown could heighten transmission. Hence, it is unclear in which direction transmission will change in response to lockdown.

We used an established, agent-based model of DENV transmission [26] to explore the impact of lockdown on dengue incidence. Our model incorporates a detailed, realistic, and spatially explicit representation of human mobility and spatiotemporal patterns of mosquito abundance. It is calibrated to dengue incidence in the city of Iquitos, Peru, and has been previously used to answer a number of questions of public health significance beyond that specific setting [27,28]. In this study, we compared the effects of initiating lockdown in different DENV transmission seasons and at different times within a transmission season. Rather than seeking to model the actual patterns of dengue in Iquitos in 2020, our study uses reconstructions of past dengue seasons, and asks the hypothetical question of what these seasons would have looked like with changes in mobility similar to those we might expect under lockdown, or with disrupted vector control campaigns. This enables us to isolate the effect of those modifications and to present general principles from our assessments. We focused on lockdown effects on (1) the incidence and spatial distribution of DENV infections, (2) local transmission by calculating household secondary attack rates, and (3) disrupted vector control campaigns. Our approach yields predictions for how lockdown could affect dengue and the mechanisms by which it could do so.

## Methods

### Model overview

A detailed explanation of all features of the model is given in the S1 Text and previous publications [26–29]. In brief, we used an established agent-based model of DENV transmission [26], with a detailed and realistic model of human movement [30], to explore how preventive measures taken against COVID-19 could affect DENV transmission. Our model is based on the city of Iquitos, in the Peruvian Amazon. Human agents in the model move around the city according to individualized movement trajectories, calibrated to data on movement patterns in Iquitos from semi-structured resident interviews [30]. The average distribution of mosquito agents in the model follows spatio-temporal estimates of abundance based on household entomological surveys [31]. The distribution of household sizes matches the demography observed in population surveys [30]. Mosquito agents also move, but much less than human agents. Each mosquito determines when to bite based on a temperature dependent gonotrophic rate parameter, and who to bite based on the body sizes of humans present at that time (see S1 Text for more details). Transmission occurs between mosquito and human agents when one agent (either mosquito or human) is susceptible and the other infectious, a blood-meal is taken by the mosquito, and the infection establishes in the susceptible agent, which occurs with a fixed probability. Transmission in the model is partially driven by time series of imported infections, which were calibrated to estimates of the time-varying, serotype-specific force of infection over an 11-year time period (S1 Fig). See Fig in Perkins *et al.* and S1 Fig in Cavany *et al.* for a visual representation of the calibration [26,27]. We do not alter these importation patterns

between simulations. In reality, importation patterns almost certainly decreased during lockdown. However, we used baseline importation patterns because epidemic timing in our model is driven by the importation time series. Moreover, our aim was to isolate the effects of mobility changes and disruptions in vector control, and for these aims the baseline importation patterns served as a valid baseline. Other differences between what actually happened in Iquitos and our simulated scenarios likely include the length and level of compliance with lockdown–these differences are discussed in more detail in the Discussion.

## Simulations

To understand the impact that lockdown could have on dengue, we simulated historical transmission under typical movement patterns and those that might be expected under lockdown. We defined lockdown as 70% of the population staying at home instead of undertaking their typical movement trajectory for that day. That is, 30% of individuals do not comply with lockdown, either because they choose not to or because they have an essential job. We explored different values of compliance in a sensitivity analysis. We chose three representative seasons from empirical data for the period 2000–2010 that span a range of seasonal dynamics, including a season with low incidence (2004–2005, "low" hereafter), a season with high incidence but no new serotype invasion (2000–2001, "high" hereafter), and a season with high incidence due to a new serotype invasion (2001–2002, invasion of DENV-3, "serotype invasion" hereafter) (Fig 1). Seasons were defined to begin on July 1 and end on June 30 [32]. In each of these scenarios, we initiated lockdown on the first of each month to explore the effect of initiating lockdown at different times of year. (Fig 1). For our baseline setting, we simulated lockdowns lasting for three months, though we also explored different values of this timespan in a second sensitivity analysis. We explored starting lockdown in different months to understand how lockdown interplays with the seasonality of dengue. Except where stated otherwise, we started lockdown on March 17, which was the date lockdown began during 2020 in Peru. This showed a moderate effect in the serotype invasion season. We simulated vector control as a city-wide

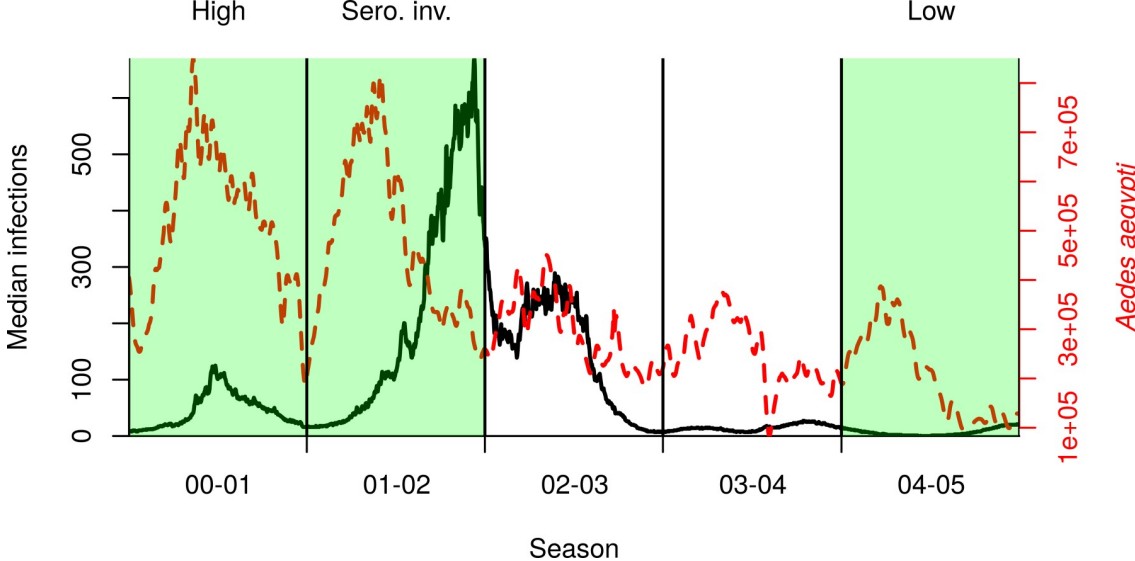

**Fig 1. Median incidence of DENV infections (black solid line) and total mosquito abundance (dashed red line) in our reconstruction of dengue virus transmission during 2000–2005 in Iquitos.** Vertical lines delineate transmission seasons, defined as beginning on July 1. Highlighted in green are three seasons in which we initiated lockdown, chosen to represent a range of possible epidemiological scenarios.

campaign in which all houses that complied (70%) were sprayed three times in a three-week period; approximately 11,000 houses per day. In baseline simulations and for the calibration, vector control did not take place. The ultra-low volume insecticide treatment had no residuality and increased the mortality rate of mosquitoes in the household by 1.5 day$^{-1}$ on the day when spraying occurred [27,33]. We ran each simulation 400 times, because increasing the number of simulations past that point did not reduce the coefficient of variation in the number of DENV infections by more than 0.1% [34].

## Analysis

For each scenario, we analyzed the distribution across simulations of the number of human DENV infections in the transmission seasons including and immediately following the initiation of lockdown, and compared this to the same period in simulations without lockdown. We calculated the following outcomes:

1. The total number of infections through space and time. Where cumulative incidence is reported in the results, it is over two seasons: the season of the lockdown and the following season. When location is shown, we assign infections to the home of the infected individual.

2. The number of unique individuals each mosquito bites during its lifetime.

3. The average secondary attack rate by location, defined as the average proportion of household contacts infected in a season, excluding the first infection in the household.

In all figures except Fig 1, we show the incidence of local infections (those infected in Iquitos), because imported infections (those infected outside Iquitos) do not differ (on average) between simulations and excluding them makes differences more visible.

The movement trajectories are calculated prior to simulating the agent-based models, and are then used as inputs to the model. By comparing the movement trajectories directly, i.e. without simulating the model, we analyzed the effect of changing these trajectories so that people spend all of their time at home. We quantified this impact by calculating the number of locations where the average number of people present was greater under lockdown These outcomes are based purely on the modified activity spaces of individuals in the model, without directly simulating movement or transmission, and are presented in the "Changing distributions of people and blood-meals section."

We undertook three sensitivity analyses: varying lockdown compliance, lockdown length, and mosquito movement probability. In all three cases, we swept across 2,000 parameter values, calculated the cumulative number of infections across two seasons, and fitted a generalized additive model (GAM) to the output using the mgcv package in R [35,36].

http://github.com/scavany/dengue_shelter_in_place

## Results

### Effect of lockdown timing on dengue incidence

The effect of the month in which lockdown was initiated varied across scenarios (Figs 2 and S2–S4). In the low and high scenarios, lockdown had little effect on the incidence of DENV infection; i.e. the timing of lockdown was not important (Fig 2). In the serotype invasion scenario, the timing of lockdown was much more important. Initiating lockdown early in the season (July–October) led to similar local two-year cumulative incidence of infections as the no-lockdown scenario; e.g., initiating in July led to 130,434 infections (95% uncertainty interval (UI): 122,674–150,995), a 0.2% decrease (Figs 2 and S5). Conversely, initiating lockdown just

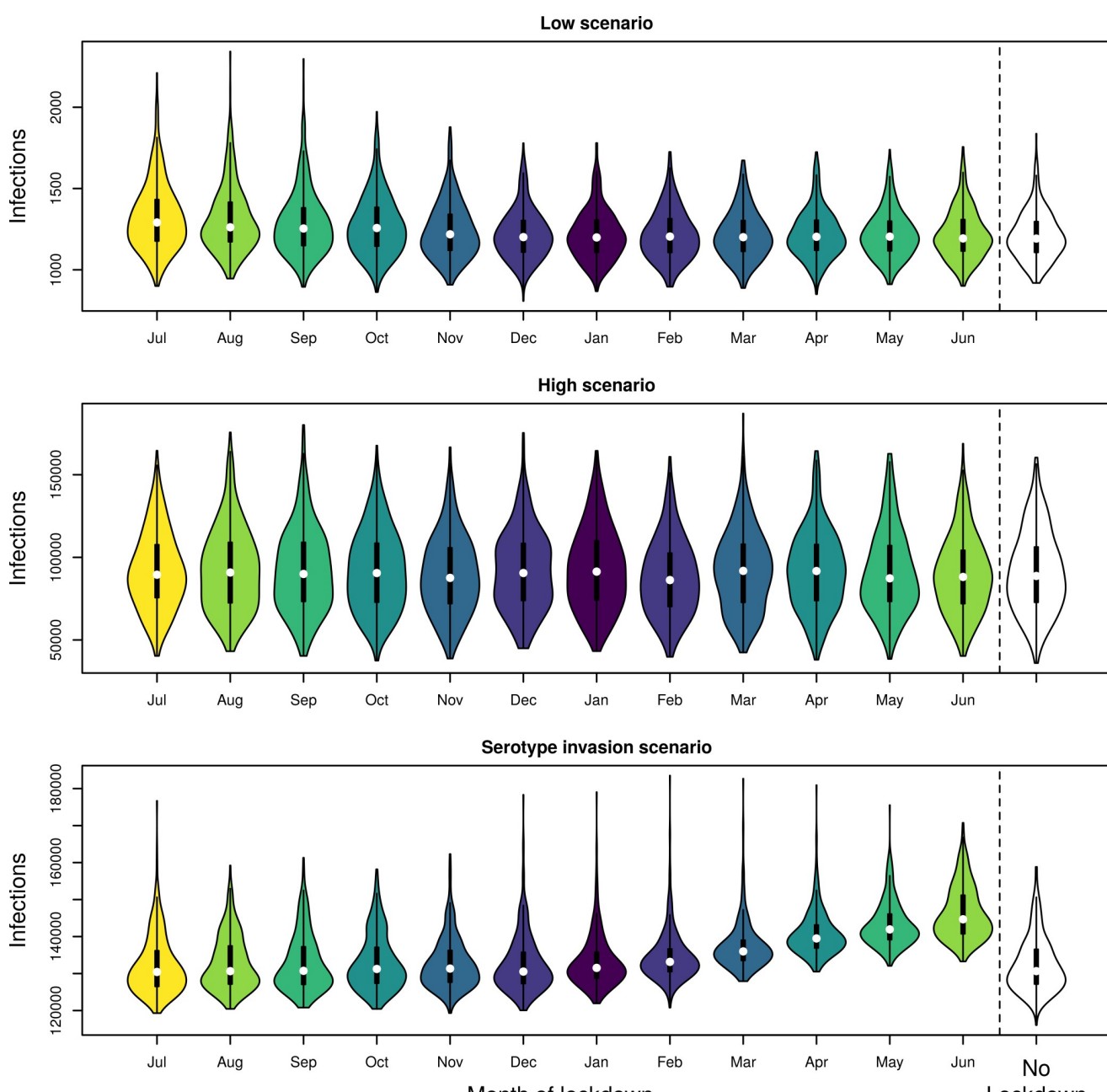

**Fig 2. Comparison of cumulative DENV infections when lockdown started on the first of the month in the shown month. Incidence was summed over both the season in which lockdown was initiated and the following season.** Different colors denote different months vs no lockdown (no color).

after the seasonal peak in infections (June) led to many more infections; e.g., 144,668 infections (95% UI: 136,812–162,403) over two years, an 11% increase. This part of the season follows the period of highest incidence, when prevalence and force of infection are at their highest.

## Spatial and locational effects of lockdown

When we initiated a three-month lockdown in mid-March in the serotype invasion scenario, lockdown changed the spatial distribution of infections; i.e. the home of the person infected

(Fig 3). Mid-March was chosen because that is when COVID-19 mobility restrictions were put in place in 2020 and because we observed a moderate effect of lockdown initiated in March in the serotype invasion scenario (Fig 2). In 20 of 35 Ministry of Health zones, infections rose, with the greatest increases in the northeast and southwest of the city. Notably, the zones with the greatest increases were those zones with the highest average *Ae. aegypti* abundance. Those with a decrease had the lowest average *Ae. aegypti* abundance (Pearson's correlation, $r = 0.946$ (95% UI: 0.923, 0.955)). The correlation between cumulative incidence and average mosquito abundance was $r = 0.925$ (95% UI: 0.893, 0.938) when no lockdown occurred, compared to $r = 0.946$ with lockdown. This indicates that spatial abundance of mosquitoes may have a slightly stronger effect on dengue incidence when human mobility is reduced. Neither human population density ($r = -0.344$) nor changes in the total number of person-days spent in each zone ($r = 0.056$) were strongly correlated with changes in incidence.

While some zones where people spend more time under lockdown were those where there was also greater mosquito abundance (e.g., in the northeast), other zones with high mosquito abundance saw reductions in the amount of time spent there (e.g., in the southwest) (Fig 3B and 3D). Notably, these southwest zones saw increases in incidence despite, in most cases, fewer person-days being spent there (Fig 3A and 3D). This is due to the fact that while fewer people were visiting this region, resulting in an overall decrease in person-hours, the people who live in this region were spending more time in homes with relatively high mosquito abundance.

The type of location where infections took place noticeably changed under lockdown. In the baseline scenario, 54.5% (95% UI: 54.3%– 55.0%) of infections occurred in the home of the infected individual. In contrast, when a lockdown occurred in mid-March in the serotype invasion season, 66.3% (95% UI: 58.9%– 70.2%) of infections occurred in the infected person's home. This had an effect on the household secondary attack rate, which increased from a mean of 0.109 (95% UI: 0.0999, 0.126) in the baseline scenario to 0.128 (95% UI: 0.119, 0.146) in the lockdown scenario, a 17% increase. In our model, lockdown had a negligible impact on the total number of mosquito bites on humans because the time when mosquitoes take blood-meals is determined by the temperature-driven gonotrophic cycle period, not the number of humans present.

## Changing distributions of people and blood-meals

According to the most recent available information from reports of past city-wide spraying campaigns (which are not publicly available), there are 92,896 buildings in Iquitos [27] (Amy Morrison & Helvio Astete, *personal communication*). Comparing typical patterns of human mobility in Iquitos with those under lockdown (assuming each person spends more time at home), we would expect 78,562 (85%) buildings to have more people inside them during lockdown. Every location that was expected to have more people under lockdown was a residential location, while all non-residential locations had fewer people, on average, under lockdown. There were 9,761 residential locations (11% of all 88,323 residential locations) that had a lower average number of people under lockdown.

Model simulations in the absence of vector control showed that the number of unique individuals bitten by a single mosquito increased under lockdown. In the serotype invasion scenario with a three-month lockdown beginning in mid-March, the number of unique individuals each mosquito bit in its lifetime rose from 2.54 to 2.64 (3.9%). In all three scenarios, the number of unique bites rose by 0.09–0.10 bites. This increase in the number of unique bites was due to heterogeneity in the number of people per location. In the baseline scenario, the Gini coefficient of the number of people in each house was 0.635, whereas under lockdown

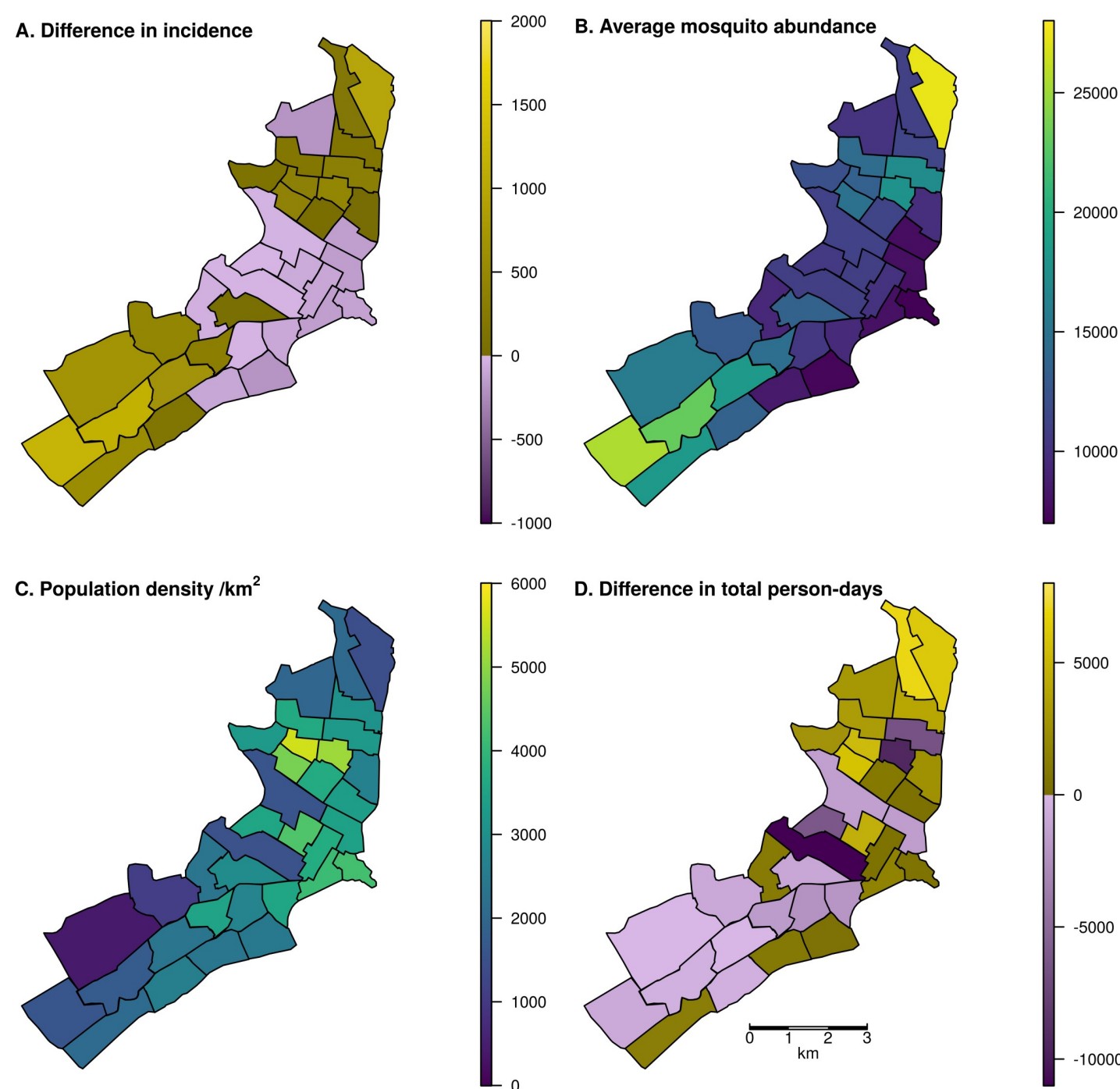

**Fig 3. Map of Iquitos, with the 35 Ministry of Health (MoH) zones delineated.** In panels A and D, yellow indicates increases and blue indicates decreases. In panels B and C, colors are a continuous scale showing the given metric. A: Spatial distribution of changes in total incident DENV infections, assigned to the home zone of the infected individual, across a two-year period including the serotype invasion and following seasons. Lockdown was initiated on March 17 in the serotype invasion season. B: Total mosquito abundance across different MoH zones, averaged across the two-year period. C: Human population density of the MoH zones. D: Difference in the total person-days spent in each zone between lockdown and baseline scenarios assuming 70% of people complied with lockdown measures. Shape files for the underlying maps can be found at github.com/scavany/dengue_shelter_in_place.

it was 0.402. As a smaller Gini coefficient implies greater homogeneity, this suggests that the number of people per house was more homogeneous under lockdown. In turn, this implies that in the baseline scenario there was more heterogeneity in the number of unique individuals available for each mosquito to bite, resulting in fewer bites on unique individuals on average.

## Importance of vector control

In addition to the effects of lockdown on incidence caused by changes in mosquito-human encounters, lockdown can affect incidence by disrupting vector control. If an early-season vector-control campaign (July or August) was interrupted by public health measures against COVID-19, the impact of lockdown and the interrupted campaign was small, but still more than double the incidence of infection immediately following lockdown (Figs 4 and S5, Jul and Aug panels). If the lockdown instead took place just prior to the seasonal peak (e.g., March), this led to a large increase in the size of the epidemic by a factor of greater than 10 at the peak of the season (Figs 4 and S5, Mar panels). In the alternative scenario in which vector control proceeded as planned during lockdown, there was a large rebound in infections the following season, due to low population immunity. That rebound could be mitigated partially by a delayed campaign following lockdown, or a city-wide campaign in the subsequent year (S6–S8 Figs).

## Sensitivity analysis

In all baseline analyses, we used values of 70% compliance with lockdown orders and a lockdown length of three months. We explored changing these values in a one-at-a-time sensitivity analysis in the serotype invasion season beginning lockdown in mid-March (S9–10 Figs). Compliance had a non-linear, non-monotonic relationship with the cumulative number of infections (S9 Fig). The cumulative number of infections peaked at slightly below 90% compliance. This indicates that while lockdown tends to increase DENV incidence, the optimal conditions for transmission require some amount of human mobility. Even if compliance were 100%, however, our results indicate that incidence would still rise compared to typical movement patterns. Longer lockdowns appeared to increase cumulative DENV incidence, though this effect saturated at around 150 days (S10 Fig).

Because human mobility was severely curtailed during lockdown and mosquito distribution patterns were correlated with changes in DENV incidence, the role of mosquito movement in transmission may have been heightened in our model. We explored this by varying the daily probability of mosquito movement in simulations with and without lockdown (Fig 5). Irrespective of whether lockdown occurred, incidence peaked when the daily probability of mosquitoes moving from a house was around 0.2; i.e. each day, each mosquito moves to a new house with probability 0.2. Regardless of lockdown, there was no transmission when mosquitoes did not move between houses, because house-level mosquito extinctions could not be replenished. As mosquito movement increased, so did the proportional change in the number of infections in lockdown compared to no lockdown (Fig 5B). This indicates that the role of mosquito movement is heightened under lockdown. If its role were the same in both situations, we would expect this ratio to remain at 1 across the range of mosquito movement probabilities.

## Discussion

We found that lockdown movement restrictions led to an increase in DENV transmission in our model output, if these restrictions occurred during a season in which incidence was high due to a new serotype invasion. Though the increase was relatively modest, the effect was most

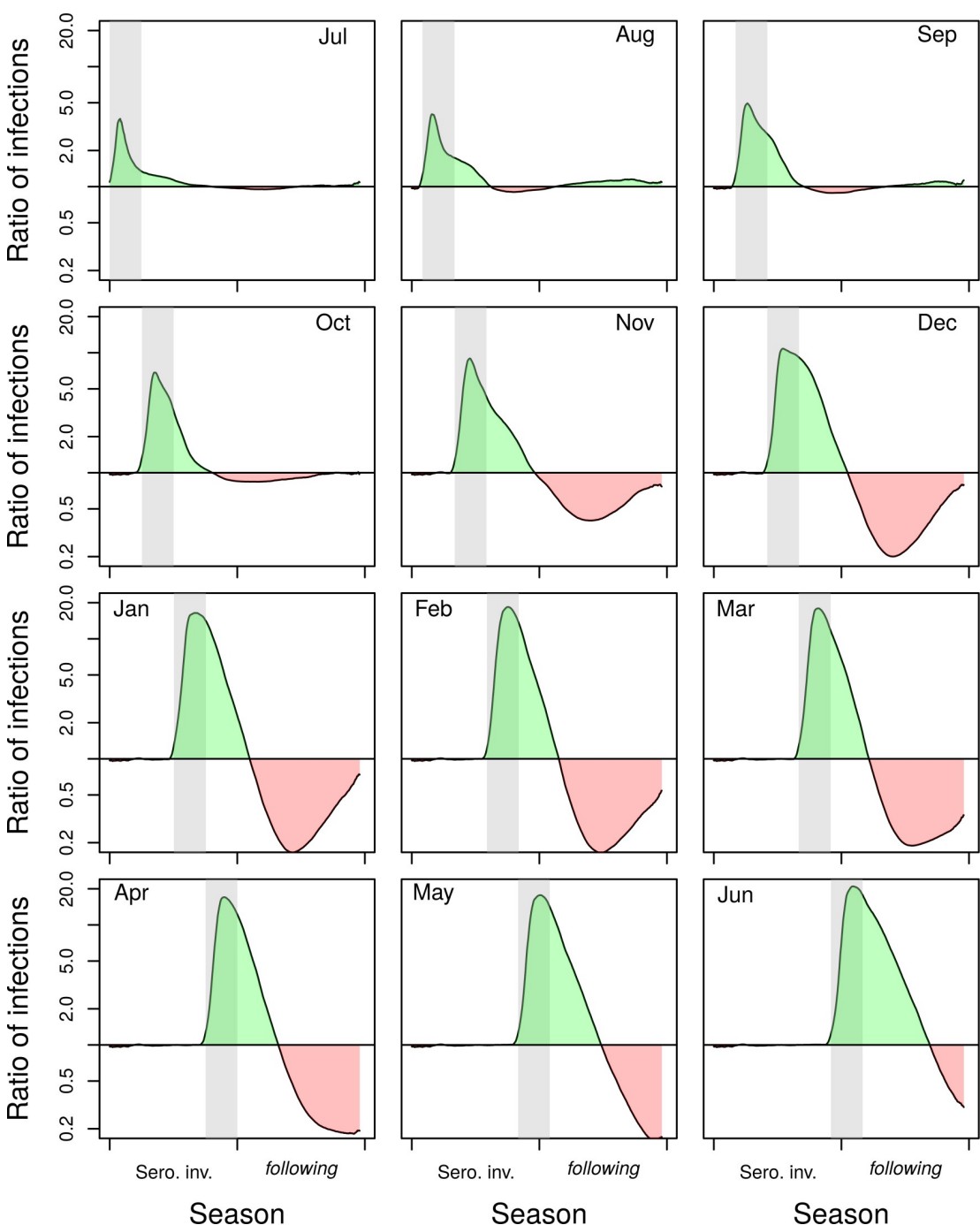

**Fig 4. Ratio of the mean number of infections under lockdown without vector control to the mean number in the baseline scenario without lockdown but with vector control.** Results are for the serotype invasion and following season, when lockdown is initiated in different months (shown in the top corner of the panel). When lockdown did not occur, there was a city-wide vector control campaign. Conversely, when lockdown did occur, there was no vector control campaign. Lockdown/vector control occurred during the gray band. Green shading indicates an increase in infections under lockdown without vector control, whereas red shading indicates a decrease.

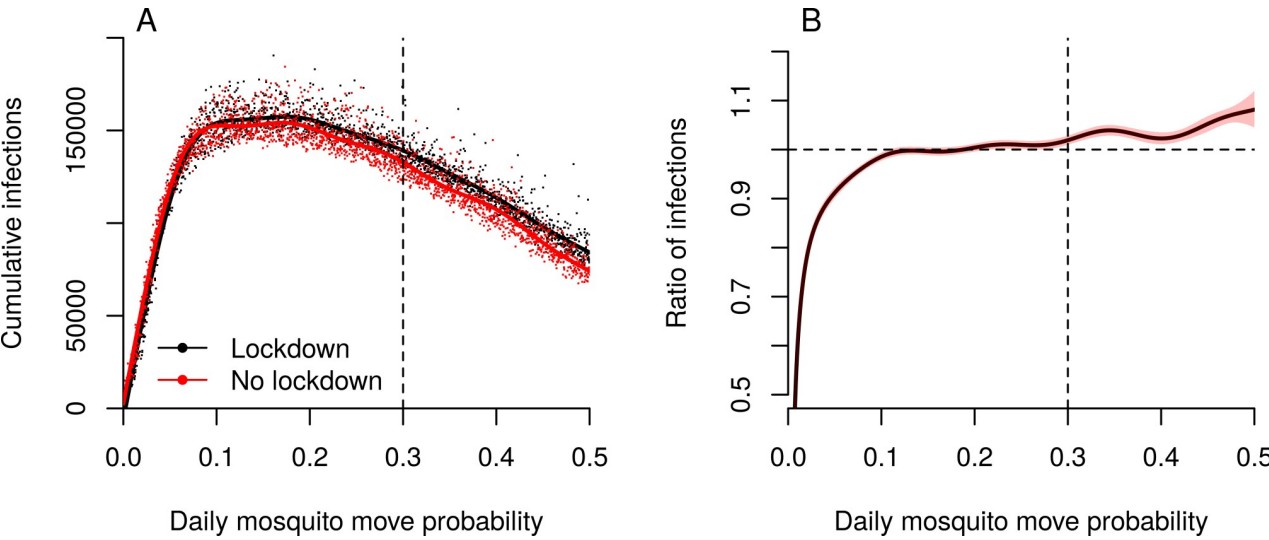

**Fig 5. Role of mosquito movement.** A. Cumulative infections for different values of daily mosquito movement probability, with and without lockdown. The dashed vertical line indicates the default value (0.3) that was used in all other simulations. B. The ratio of the average number of infections in a lockdown scenario to the number without lockdown. The dashed horizontal line represents when these two situations are the same. If the solid line is above the dashed horizontal line, lockdown resulted in more cumulative infections than when there was no lockdown. The dashed vertical line indicates the default value that was used in all other simulations.

pronounced when lockdown occurred during periods of time when the prevalence of infection, and hence the force of infection, was highest. In this study, that scenario occurred at the end of a novel serotype invasion season, following several months of heightened transmission. Lockdown had a more pronounced effect in changing the locations where transmission occurred and the spatial distribution of infections. Specifically, more infections occurred in people's homes than at other types of locations, which increased the household secondary attack rate by 17%. This meant that more infections occurred where mosquito abundance was highest, amplifying hyperlocal transmission due to incomplete compliance with lockdown and mosquito movement among nearby houses. Mosquito movement between houses seems particularly important to enable some inter-household transmission in the context of reduced human mobility. If mosquitoes moved less, the effect of lockdown would in fact be to reduce transmission overall. When we considered the effect of potential interruptions to vector control, lockdown led to much greater increases in DENV transmission.

Our results identified three factors contributing to the direct effect of lockdown on DENV transmission apart from interruption of vector control. First, lockdown causes infections to become more concentrated in locations where mosquito abundance is highest, facilitating more transmission than might occur otherwise. Second, people spending more time at home causes an increase in the household secondary attack rate. Under non-lockdown circumstances, uninfected household members spend more time outside the home and thereby reduce their exposure. Third, lockdown results in a more homogeneous distribution of person-hours across locations, which leads to an increase in the number of unique individuals that each mosquito bites because there are fewer mosquitoes with no-one to bite and many mosquitoes have more options of who to bite. This increases the chance that a mosquito becomes infected within its lifetime, as well as the chance that a mosquito gets infected by one person and later bites and infects a different person.

These direct effects of lockdown are likely to be greatest in settings where transmission predominantly occurs in homes and where household-level mosquito abundance is typically high.

In Iquitos, this assumption is supported both by pupal surveys of *Aedes aegypti* mosquitoes in non-residential locations [37] and epidemiological investigations of contact-site clusters [23,38]. Rather than to model the interaction between COVID-19 and dengue as it unfolded in Iquitos, however, our goal was to use a model previously developed for Iquitos to address a general question about the effect of lockdown on dengue. We used hypothetical scenarios with altered mobility patterns and vector control campaigns, which were not intended to directly represent the reality of what happened in Iquitos or any other city in 2020, but to gain mechanistic understanding of the effect of these type of mobility changes. Even so, it is worth noting that Iquitos did experience a large COVID-19 epidemic beginning in March 2020, which severely strained health services there [39]. Just before then, the city experienced a relatively large dengue epidemic in December 2019 through March 2020 [40]. Extensive disruption of health services thereafter for illnesses other than COVID-19 makes it difficult to know what the course of that dengue epidemic was once COVID-19 arrived [39]. Given how large the COVID-19 epidemic in Iquitos appears to have been [41], compliance with lockdown there may have been low. On the other hand, Google mobility data from Maynas province shows a 30% increase from baseline in household mobility and a 60% decrease from baseline in workplace mobility during April and May, suggesting substantial changes in mobility patterns [42]. Although low compliance with lockdown would reduce the direct effects of lockdown on dengue that we demonstrated with our model, reductions to vector control services may have had effects on dengue incidence that went unnoticed by surveillance and that could have implications for the next dengue transmission season. If we were to use our model to attempt to reconstruct what actually happened in Iquitos, we need data from serological surveys and mosquito household surveys from 2010 up to 2021, alongside detailed mobility data for the city. Once these former data become available, such a study could prove a valuable contribution.

Globally, there have been reports of both rising [7,43,44] and falling [45,46] dengue incidence since the COVID-19 pandemic began [47]. A study from Thailand was able to associate increases in dengue incidence with interventions against COVID-19 [48], consistent with our findings. The observational nature of that study did not allow for the mechanisms behind that association to be understood, but the authors hypothesized that it may have been due to heightened exposure to vectors while people spent more time in their homes [48]. Our analysis adds value by testing that hypothesis through simulation experiments and elucidating how the strength of those effects is modulated by other factors, such as spatial heterogeneity in mosquito abundance, the spatial scale of mosquito movement, compliance with lockdown, and the seasonal timing of lockdown. Our study is also inline with a statistical analysis from Brazil finding a positive association of mobility restrictions with dengue cases 20 days later [49], and is similar to the finding that reductions in mobility due to the effects of fever can increase transmission [50]. Conversely, a study from Sri Lanka found a decreased risk of dengue during lockdown [51].

A factor that could be important in modulating effects of lockdown on dengue is the extent to which DENV transmission occurs at other types of locations, such as schools [52,53]. In settings where schools or other non-residential locations are important for DENV transmission, lockdown measures could have a qualitatively different effect by reducing the number of people in those high-risk locations. Nonetheless, homes and their vicinity remain a key site of transmission in many settings, and we expect that our finding that lockdown further increases transmission in such locations will be robust [54–59]. Spatio-temporal heterogeneity in compliance with lockdown may also impact dengue transmission. While we modeled some heterogeneity by randomly choosing who will comply with lockdown, there is no spatial or temporal structure to this aspect of our approach. If certain districts within a city have lower compliance with lockdown, there may be a reduced effect of lockdown in those districts.

The scale of the non-pharmaceutical interventions undertaken to combat the COVID-19 epidemic have likely impacted a number of other diseases, either directly or indirectly. Some of these effects may be positive. For instance, due to the shared route of transmission, interventions against COVID-19 impact influenza in a similar way, and likely contributed to very low flu seasons during the Southern Hemisphere winter and in Hong Kong [60,61]. On the other hand, disruption of key services and reductions in care-seeking behavior are projected to have negative effects on the burden of TB, malaria, HIV/AIDS, and a range of vaccine-preventable diseases [2,3,62–65]. Our results align more closely with the latter pattern, showing a potentially large negative effect if dengue vector control campaigns are interrupted. To mitigate this, public health authorities could encourage or assist people to spray their own homes, by providing them with self-use insecticide treatments [66]. This would mitigate the impact of reduced vector control activities and reduce the impact of changes in mobility on dengue transmission.

A significant strength of our study is our spatially explicit treatment of human mobility, which allowed us to isolate the effect of lockdown in ways that a simpler model could not. Moreover, our model's direct inclusion of mosquito movement and individual biting behavior allowed us to understand the changing role of these factors in DENV transmission when movement restrictions were imposed. Our study also has at least five limitations. First, it is difficult to know the exact response people made to lockdown measures, such as the level of compliance and how the nature of their movements changed. Our sensitivity analysis of lockdown compliance and duration found that reduced compliance linearly decreased the change in incidence due to lockdown. Second, we assumed that mosquito behavior was unaffected by changes in human mobility. Lockdown has been associated with increased vector indices in India [67]. Third, we did not assess the impact of changing DENV importation patterns into Iquitos. Because our model predicts the biggest impact of lockdown is late in the season, a time when the epidemic is predominantly driven by local transmission, we would not expect changes in imported infections to qualitatively affect this result. Changing importations may, however, have a significant impact if they prevent an imported infection from seeding a new outbreak, particularly if the introduced virus was a new serotype. Fourth, we did not assess impact in terms of severe disease; e.g., dengue hemorrhagic fever (DHF). We made this decision because of severe dengue's dependence on the local immunity profile and circulating serotype, which would mean DHF results would be difficult to generalize. Our model was also calibrated to a statistical reconstruction of incidence of infection [25] rather than disease. Nonetheless, increased incidence of DENV infection would, all else being equal, be expected to translate to higher rates of severe disease; a very concerning situation in the context of already strained health systems due to COVID-19. Fifth, in order to simplify the analysis we did not incorporate structured spatial heterogeneity in compliance. While we do model heterogeneity in compliance between individuals, structured differences between regions may lead to effects not captured in our analysis.

Our findings illustrate why, during a syndemic, public health officials must consider the implications of an action to prevent one disease on other concurrent diseases [68]. Thus, a holistic approach to infectious disease mitigation is needed. Research and policy efforts should focus on ways to retain the positive effects of lockdown on diseases like COVID-19, influenza, and pneumonia while mitigating the negative effects on dengue, malaria, and TB. Vector control activities which people can do themselves in their own homes, such handheld over-the-counter insecticide treatments, should be encouraged and supported during lockdown. Whenever there is risk of DENV transmission, efforts must be made to avoid disrupting effective control practises and provide carefully planned alternative means of making interventions possible. More communication, creativity, and inter-sectorial collaboration will be needed to

ensure the continuation of meaningful interventions than to rely solely on providing vector control staff with personal protective equipment to carry out existing forms of control [18].

## Supporting information

**S1 Text. Supplementary Methods.**
(PDF)

**S1 Fig. Monthly, serotype-specific incidence of infection per capita, as estimated by Reiner et al. [25] (gray bands) and as reproduced by our calibrated model (colored bands).** Taken from [27]
(TIF)

**S2 Fig. Time series of local DENV infections across the low-transmission and following seasons when lockdown was initiated on the first of the month (dashed line) in the low season and lasted for two months (ending at the dotted line).** Shaded regions are the interquartile range. Shading in gray is where these regions overlap.
(TIF)

**S3 Fig. Time series of local DENV infections across the high and serotype invasion seasons when lockdown was initiated on the first of the month (dashed line) in the high season and lasts for two months (ending at the dotted line).** Shaded regions are the interquartile range. Shading in gray is where these regions overlap.
(TIF)

**S4 Fig. Time series of local DENV infections across the serotype invasion and following seasons when lockdown was initiated on the first of the month (dashed line) in the serotype invasion season and lasts for two months (ending at the dotted line).** Shaded regions are the interquartile range. Shading in gray is where these regions overlap.
(TIF)

**S5 Fig. Time series of local DENV infections in the serotype invasion scenario, comparing lockdown without vector control (purple) to no lockdown with vector control (green).** Lockdown and the city-wide vector control campaign began at the dashed line. Lockdown lasted three months (ending at the dotted line). The vector control campaign lasted three weeks. Shaded regions are the interquartile range. Shading in gray is where these regions overlap.
(TIF)

**S6 Fig. Time series of local DENV infections in the serotype invasion scenario, comparing lockdown without vector control (purple) to no lockdown with vector control (green).** Both lockdown and the city-wide vector control campaign began at the dashed line in the first season. In the following season, vector control occurred in both simulations and lockdown did not occur in either simulation. Lockdown lasted three months (ending at the dotted line). The vector control campaign lasted three weeks. Shaded regions are the interquartile range. Shading in gray is where these regions overlap.
(TIF)

**S7 Fig. Ratio of the mean number of infections under lockdown to the mean number in the baseline scenario without lockdown in the serotype invasion and following season.** Lockdown began at the vertical dashed line, and ended at the dotted line. In the following season, vector control occurred in both simulations and lockdown did not occur in either simulation.
(TIF)

**S8 Fig. Time series of local dengue infections in the serotype invasion scenario, comparing vector control without lockdown (purple) to lockdown with a vector control campaign which begins as soon as lockdown ends (green).** Lockdown lasted three months, starting at the dashed line and ending at the dotted line. The vector control campaign lasted three weeks, beginning at the dashed line (purple) or the dotted line (green). Shaded regions are the inter-quartile range. Shading in gray is where these regions overlap.
(TIF)

**S9 Fig. Effect of changing lockdown compliance on the proportional change in the cumulative number of infections across two consecutive seasons.** The vertical dashed line shows baseline compliance (70%). The horizontal dashed line shows when there is no effect of lock-down.
(TIF)

**S10 Fig. Effect of changing lockdown length on the proportional change in the cumulative number of infections across two seasons.** The vertical dashed line shows the baseline length (three months). The horizontal dashed line shows when there is no effect of lockdown.
(TIF)

## Acknowledgments

We thank Amy Morrison (University of California, Davis) for helpful discussion and commentary on the manuscript. We also thank the Proyecto Dengue team in Iquitos for performing research over many years that informed the development of our model. This research made extensive use of computing resources provided by the Center for Research Computing at the University of Notre Dame.

## Author Contributions

**Conceptualization:** Sean M. Cavany, Guido España, Gonzalo M. Vazquez-Prokopec, Thomas W. Scott, T Alex Perkins.

**Data curation:** Sean M. Cavany.

**Formal analysis:** Sean M. Cavany, Guido España, Gonzalo M. Vazquez-Prokopec, Thomas W. Scott, T Alex Perkins.

**Funding acquisition:** Thomas W. Scott, T Alex Perkins.

**Investigation:** Sean M. Cavany, Guido España, Gonzalo M. Vazquez-Prokopec, Thomas W. Scott, T Alex Perkins.

**Methodology:** Sean M. Cavany, Gonzalo M. Vazquez-Prokopec, T Alex Perkins.

**Project administration:** Sean M. Cavany.

**Resources:** Sean M. Cavany.

**Software:** Sean M. Cavany, Guido España, T Alex Perkins.

**Supervision:** Thomas W. Scott, T Alex Perkins.

**Validation:** Sean M. Cavany.

**Visualization:** Sean M. Cavany.

**Writing – original draft:** Sean M. Cavany.

**Writing – review & editing:** Sean M. Cavany, Guido España, Gonzalo M. Vazquez-Prokopec, Thomas W. Scott, T Alex Perkins.

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
