## [Decision Letter · Decision Letter 0]

19 Jan 2021

Dear Dr. Cavany,

Thank you very much for submitting your manuscript "The impacts of COVID-19 mitigation on dengue virus transmission: a modeling study" for consideration at PLOS Neglected Tropical Diseases. As with all papers reviewed by the journal, your manuscript was reviewed by members of the editorial board and by several independent reviewers. In light of the reviews (below this email), we would like to invite the resubmission of a significantly-revised version that takes into account the reviewers' comments. 

We cannot make any decision about publication until we have seen the revised manuscript and your response to the reviewers' comments. Your revised manuscript is also likely to be sent to reviewers for further evaluation.

Sincerely,

Rebecca C Christofferson

Associate Editor

Jeremy Camp

Deputy Editor

Editors' comments:

I have read the comments and would ask that you pay particular attention to justifying the timing of the lockdown, as well as provide justification/ referencing for some of the parameter values. Additionally, the language in the discussion should be softened the reflect the assumptions and repercussions thereof of the model.

Reviewer's Responses to Questions

**Key Review Criteria Required for Acceptance?**

**Methods**

-Are the objectives of the study clearly articulated with a clear testable hypothesis stated?

-Is the study design appropriate to address the stated objectives?

-Is the population clearly described and appropriate for the hypothesis being tested?

-Is the sample size sufficient to ensure adequate power to address the hypothesis being tested?

-Were correct statistical analysis used to support conclusions?

-Are there concerns about ethical or regulatory requirements being met?

Reviewer #1: Generally the objectives and methods are clear. A few questions below:

1. lines 103-104 How could this calibration effect the results of this study? for example, have movement within and into Iquitos and/or populations, buildings, etc. change significantly since the early 2000/s, especially during the pandemic? How would this affect the outcomes?

2. lines 108-109 Does this mean you are assuming that 30% of the people have essential jobs? Or that 70% of the people without essential jobs are staying home? Also, this doesn't necessarily need to be included in this study, but it are the essential workers more likely to come from certain parts of the city or higher incidence neighborhoods?

3. lines 121-123 It's not always clear when vector control is applied versus not. For example, did the original calibration include vector control? 

4. Figure 1, is there a comparison with the reported case counts and the simulated data?

5. Line 144 What does "without simulating the full model" mean?

6. Line 148 why is the unique number of visitors greater under lockdown? wouldn't there be the same household members at home without extra people visiting?

7. Line 152 I assume that "compliance" is referring to lockdown and not the vector control compliance referenced earlier?

Reviewer #2: The hypothesis of the study is clear and the study design is appropiate.

Methodology is appropiate but it is necessary to search for other papers to fully understand it.

Minor comment: In the line 98-99 is stated that “Each mosquito determines when to bite based on a temperature dependent rate parameter”, I would like to know how other variables that are known to influence feeding like gonotrophic cycle or body size are involved in this part. It is also not very clear in the supplementary material.

Reviewer #3: See general comments. Major issues with study design

**Results**

-Does the analysis presented match the analysis plan?

-Are the results clearly and completely presented?

-Are the figures (Tables, Images) of sufficient quality for clarity?

Reviewer #1: The results match and presentation is generally clear. A few questions below.

1. Lines 163-164 Interesting that one month difference in timing makes such a big difference in results. Is this June lockdown referred to here a month before the July lockdown or 11 months later in the simulation? I guess more specifically, is it right before the introduction of a new serotype or 11 months after? From looking at the first figure, June is normally low incidence but happened to be particularly high because of the timing of the serotype invasion that season which allowed transmission even during a normally lower mosquito density because of the high susceptibility of the human hosts. So it's not so much the particular month as it is whether or not the lockdown is happening when incidence is over some kind of high threshold. Is that a correct interpretation?

2. Figure 2 says you are including two years worth of incidence in the plot but this wasn't clear from the Methods section or in other descriptions--whether or not you are considering the one season or multiple in the results.

3. Line 173 "spatial distribution" Are you referring to the location where infectious bite occurred or the home of the person infected? (This applies to the spatial maps of infection as well)

4. Lines 186-189 Not sure that it makes sense to have higher incidence with fewer people-hours. More explanation would be helpful. Were incidence increases were in people residing there? Or people infected while visiting there?

5. Lines 208-214 These numbers are confusing. It says 78,562 buildings have more people inside them but then says 9,761 residential locations had higher numbers of people during lockdown

6. Lines 217-225 It's not clear when vector control is occurring or not in these descriptions.

7. Figure 5. How does the movement rate determined in the original calibration compare to the numbers shown in the plot? is there reason to think that mosquito movement increased or is this more of a "what-if" scenario?

8. Lines 293-295 Why does more homogeneity result in fewer unique bites? This seems counter-intuitive

Reviewer #2: Results do match analysis plan.

Some comments: 

Lines 179-181: “The correlation between cumulative incidence and average mosquito abundance was r = 0.925 when no lockdown occurred, compared to r = 0.946 with lockdown. This indicates that spatial abundance of mosquitoes becomes a stronger determinant of dengue incidence when human mobility is reduced.” It doesn’t seems enough difference to indicate the latter statement.

I suggest to include values related to statistical significance when spearman correlation coefficient is indicated (p-value for example). Correlation coefficient indicates strength and direction of the correlation but it is not necessarily statistically significant.

Line 213: it is not clear where the value of “11% of all residential locations” comes from. Seems a little lower which, in turn, can tend to underestimate the blood-meal estimates.

Reviewer #3: See general comments below

**Conclusions**

-Are the conclusions supported by the data presented?

-Are the limitations of analysis clearly described?

-Do the authors discuss how these data can be helpful to advance our understanding of the topic under study?

-Is public health relevance addressed?

Reviewer #1: The conclusions are robust and testable by data in the future. The limitations are stated in the manuscript and the effects of COVID-19 mitigations on other infectious diseases are of interest to a wide range of researchers and of high public health relevance.

Reviewer #2: Conclusions are well supported. Strengths and limitations are well described.

I suggest a better and more complete explanation about why there is a markedly increase in infections at homes. It is understood that people spend more time at homes but still it is important the movement of people for the arrival of the virus into the house. There are reports where staying at home during the day was protective against the disease (for example Cordeiro, R., Donalisio, M.R., Andrade, V.R. et al. Spatial distribution of the risk of dengue fever in southeast Brazil, 2006-2007. BMC Public Health 11, 355 (2011). https://doi.org/10.1186/1471-2458-11-355). This is also important at the light of poor explanation about changes of mosquito behavior.

Reviewer #3: The conclusions will need to updated after major revisions of the study design. See general comments below

**Editorial and Data Presentation Modifications?**

Reviewer #1: (No Response)

Reviewer #2: Line 163: possibly the fig 3 is wrongly referenced.

Line 262: It is better if you explain the unit of the number 0.2

Reviewer #3: See general comments below

**Summary and General Comments**

Reviewer #1: (No Response)

Reviewer #2: This is a good piece of work explaining how current issues influence old issues. It is well written and the most speculative part (such as the amount of compliance of the lockdown) are well addressed.

Reviewer #3: This study seeks to understand the impact of the COVI-19 pandemic on dengue transmission. This is a scenario based study looking at the the impact of lockdown via decrese human mobility have on DENV burden. This is timely and a major short/ long term concern in a pandemic context. However I have major concerns with the syudy design and thus the uselfullness of the simulations presented. 

Specific comments:

Why simulate 70% lockdown as ad hoc? Even though you do sensitivity analysis around this value, Are the reduction in movement realistic reductions? Why not use data driven reductions? Also we know that individual response to lockdown is heterogenous. It would be good to see actual movement data for the region used, these data while not easily available before the pandemic, are now widely available (eg., facebook mobility data, Apple, google). Reference 30 is from 2014 as such would only serve as baseline movement pre pandemic. These crude homogenous decreases end up being over -under estimations of the actuall patterns of movement. 

In Peru About 70% of the employed population work in the informal sector, which is one of the highest rates in Latin America. These jobs are by their nature unpredictable and often in environments making social distancing difficult. Peruvians who went out to work had to use public transport, and to sell goods in very crowded markets. Even during lockdown. So your 70% baseline compliance seems to be a gross over estimation. 

The latest National Household Survey suggests 11.8% of poor households in Peru live in overcrowded homes. So I wonder if your household transmission may nee dto be reasssed?

Is there evidence for the decreased vector control scenario? When we considered that lockdown measures could disrupt regular, city-wide vector control campaigns, the increase in incidence was more pronounced than with lockdown alone, especially if lockdown occurred at the optimal time for vector control.

I don’t understand the point of this scenario: Initiating lockdown early in the season (July – October) what July is this? July 2020? They started lockdown in March that lasted until June. This scenario makes no sense. If it is July 2019 makes even less sense. 

Also heterogeneous startegies between regions in Peru outside of iquitos? Why not add a few mor regions?

 Also wouldn’t stay at home scanerios be accompanied by increased portectionin the homes? Why assume nothing changes?

If there is evidence of reduction on vector control campains (I dindt find any) Should perhaps instead of a all or none control campaign, a spectrum of coverage be explored?

PLOS authors have the option to publish the peer review history of their article (what does this mean?). If published, this will include your full peer review and any attached files.

Reviewer #1: No

Reviewer #2: No

Reviewer #3: No
---

## [Decision Letter · Decision Letter 1]

27 May 2021

Dear Dr. Cavany,

Thank you very much for submitting your manuscript "The impacts of COVID-19 mitigation on dengue virus transmission: a modeling study" for consideration at PLOS Neglected Tropical Diseases. As with all papers reviewed by the journal, your manuscript was reviewed by members of the editorial board and by several independent reviewers. The reviewers appreciated the attention to an important topic. Based on the reviews, we are likely to accept this manuscript for publication, providing that you modify the manuscript according to the review recommendations. 

Hello, in addition to the reviewers, both handling editors have reviewed the manuscript, comments, and revisions. We feel that the manuscript is meritorious, but there is still some minor comments to address. Specifically, we would suggest that it be explicitly stated in both the introduction and discussion that while the manuscript is informed by Iquitos data, this is a generalized look at hypotheticals. We further suggest that major differences in the modeled scenarios vs. what actually happened in Iquitos be addressed somewhere in the manuscript. - RCC

Please consider modifying the title so as not to overstate the association with COVID-19, explicitly, but rather to reflect that the model assesses a hypothetical lockdown scenario (e.g., in response to the COVID-19 pandemic). - JVC

Sincerely,

Rebecca C Christofferson

Associate Editor

Jeremy V. Camp

Deputy Editor

Hello, in addition to the reviewers, both handling editors have reviewed the manuscript, comments, and revisions. We feel that the manuscript is meritorious, but there is still some minor comments to address. Specifically, we would suggest that it be explicitly stated in both the introduction and discussion that while the manuscript is informed by Iquitos data, this is a generalized look at hypotheticals. We further suggest that major differences in the modeled scenarios vs. what actually happened in Iquitos be addressed somewhere in the manuscript.

Reviewer's Responses to Questions

**Key Review Criteria Required for Acceptance?**

**Methods**

-Are the objectives of the study clearly articulated with a clear testable hypothesis stated?

-Is the study design appropriate to address the stated objectives?

-Is the population clearly described and appropriate for the hypothesis being tested?

-Is the sample size sufficient to ensure adequate power to address the hypothesis being tested?

-Were correct statistical analysis used to support conclusions?

-Are there concerns about ethical or regulatory requirements being met?

Reviewer #1: addressed my questions and methods adhere to criteria

Reviewer #2: The hypothesis is clear and the study design are adequated. 

It is evident that the city of Iquitos is well known and that has been characterized for long time. This can be seen through the developed model on this city. It is necessary to read other papers to fully understand the model, but the focus of the paper is not the development of the model, so it is not a big problem.

Reviewer #3: See general comments

**Results**

-Does the analysis presented match the analysis plan?

-Are the results clearly and completely presented?

-Are the figures (Tables, Images) of sufficient quality for clarity?

Reviewer #1: yes to all criteria

Reviewer #2: Results do match the analysis plan.

Results are now more clearly presented.

Reviewer #3: See general comments

**Conclusions**

-Are the conclusions supported by the data presented?

-Are the limitations of analysis clearly described?

-Do the authors discuss how these data can be helpful to advance our understanding of the topic under study?

-Is public health relevance addressed?

Reviewer #1: yes to all criteria

Reviewer #2: Conclusions are well supported. Strengths and limitations are well described.

Just a little concern: Most of simulations were done on the “serotype invasion” season, which were the one where lockdown had an effect on incidence. There is no evidence that a serotype invaded Iquitos during the 2020 (or it is not stated in the manuscript). 

At the end of the manuscript, the idea that lockdown has an effect on dengue incidence remains. So, conclusions must specify that the effects seen on most of simulations along this study apply in that especific season. For the other two scenarios, probably the most likely to have occurred, no effect was seen due to lockdown, according to the first simulations (and Fig 1). I know that in each experiment it is stated the season where experiments were runned, but I would like to suggest that be sure that correct idea remains at the end of the document.

Reviewer #3: See general comments

**Editorial and Data Presentation Modifications?**

Reviewer #1: (No Response)

Reviewer #2: None

Reviewer #3: See general comments

**Summary and General Comments**

Reviewer #1: The authors did a good job of addressing my questions and comments.

Reviewer #2: This is a good piece of work and is well written. It is appropriate for the current times and the situation. It is a well use of a previously developed model.

Reviewer #3: The authors study the impact of COVID-19 measure on dengue transmission using an ABM with mobility and vector control. 

While the simulation study is potentially interesting, I still have major issues with the assumptions in the model particularly the choices for lockdown stringency and duration.

Below I make more specific comments on my issues.

Assumptions of the model

I still have major issues with the methodology under the lockdown.

The authors mention that “Transmission in the model is partially driven by time series of imported infections, which were calibrated to estimates of the time-varying, serotype-specific force of infection over an 11-year time period.” Is this still valid in the lockdown?

“See Fig 3 in Perkins et al. and S1 Fig in Cavany et al. for a visual representation of the calibration” Why not just have the figure in SI? This seems a little lazy…

“Movements… those that might be expected under lockdown” Was this validated with any type of mobility data?

I had this same comment the first round of submission: How did you arrive at the 70% staying home? This is awfully high? In Peru About 70% of the employed population work in the informal sector, which is one of the highest rates in Latin America. These jobs are by their nature unpredictable and often in environments making social distancing difficult. Peruvians who went out to work had to use public transport, and to sell goods in very crowded markets. Even during lockdown. So your 70% baseline compliance seems to be a gross overestimation. 

There were some 2-week period lockdowns enforced with the last one in January 2021, but no record of such a stringent number. 

I don’t think this paper shows useful simulations…. The choice of seasons is weird, “this back in time” is slightly pointless as these seasons were affected by mobility, weather and other factors of those years.

While I appreciate the sensitivity analysis that I had requested, I still think that the baseline should be the actual length and the actual dates of lockdowns. Also why not use information of dengue in this season?

Why is the lockdown 3 months as the baseline? This is not what happened? 

Line 219: “According to the most recent available information, there are 92,896 buildings in Iquitos.” Why not cite where this information came from?

I’m also confused as to why under lockdown the number of people per house is more homogeneous. Shouldn’t we except this to actually be the opposite? Since these regions vary widely in terms of socio-demography. The latest National Household Survey suggests 11.8% of poor households in Peru live in overcrowded homes. So I wonder if your household transmission may need to be reassessed?

Line 257: “In all baseline analyses, we used values of 70% compliance with lockdown orders and a lockdown length of three months.” I still don’t understand why this is the baseline. This is not realistic.

Because I don’t understand the baseline and the choice of season, I have trouble believing the results.

Further, there is high heterogeny is compliance between regions and so this blanket compliance seem unrealistic. The authors argue that a major strength is the study is the spatial component, yet I don’t see any sensible calibration in this regard. On the contrary.

Im pasting comments from my first round review that I feel still haven’t been address: 

Why not use data driven reductions? Also we know that individual response to lockdown is heterogeneous. It would be good to see actual movement data for the region used, these data, while not easily available before the pandemic, are now widely available (eg., facebook mobility data, Apple, google). Reference 30 is from 2014 as such would only serve as baseline movement pre pandemic. These crude homogenous decreases end up being over -under estimations of the actual patterns of movement. 

Last time I also asked If there is evidence of reduction on vector control campaigns (I didn't find any) Should perhaps instead of all or none control campaigns, a spectrum of coverage be explored?

PLOS authors have the option to publish the peer review history of their article (what does this mean?). If published, this will include your full peer review and any attached files.

Reviewer #1: No

Reviewer #2: No

Reviewer #3: No

Figure Files:

Data Requirements:

Reproducibility:

References

---

## [Editor Report · Decision Letter 2]

28 Jun 2021

Dear Dr. Cavany,

We are pleased to inform you that your manuscript 'Pandemic-associated mobility restrictions could cause increases in dengue virus transmission' has been provisionally accepted for publication in PLOS Neglected Tropical Diseases.

Best regards,

Rebecca C Christofferson

Associate Editor

Jeremy Camp

Deputy Editor

---

## [Editor Report · Acceptance letter]

4 Aug 2021

Dear Dr. Cavany,

We are delighted to inform you that your manuscript, "Pandemic-associated mobility restrictions could cause increases in dengue virus transmission," has been formally accepted for publication in PLOS Neglected Tropical Diseases.

Best regards,

Shaden Kamhawi

co-Editor-in-Chief

Paul Brindley

co-Editor-in-Chief
